# Organic–Inorganic Hybridization of Silkworm Cocoon Filaments Using Nano Pastes of Silica–Phosphate–M (M = Cu, Fe, or Al)

**DOI:** 10.3390/nano14211697

**Published:** 2024-10-23

**Authors:** I Wayan Karyasa, Enike Dwi Kusumawati, Retno Agustarini, Lincah Andadari, Herman Sari

**Affiliations:** 1Department of Chemistry, Faculty of Mathematic and Natural Sciences, Universitas Pendidikan Ganesha, Buleleng 81116, Bali, Indonesia; 2Faculty of Animal Husbandry, Universitas PGRI Kanjuruhan Malang, Malang 65148, Jawa Timur, Indonesia; enike@unikama.ac.id; 3Center for Applied Zoology, Badan Riset dan Inovasi Nasional, Jakarta 10340, Jakarta Pusat, Indonesia; retno.agustarini@gmail.com (R.A.); a.lincah@yahoo.com (L.A.); hermansari1234@gmail.com (H.S.)

**Keywords:** siliconization, silk fiber, nano pastes, natural dyeing, antibacterial

## Abstract

Inorganic–organic hybrid biomaterials have recently attracted much attention because of their widespread use. Silkworm cocoon filaments resulting from sericulture as prospective nanobiomaterials need to be improved, and their properties need to be used for broader purposes. This study was aimed at investigating methods for siliconization of silkworm cocoon filaments and characterizing their cocoon filament properties in terms of their yarn quality, natural dyeing, and antibacterial properties. Three methods of hybridization processes were used in this experiment, namely, in situ natural dyeing of silk yarns while silk filaments were spined, feed engineering through spraying the mulberry leaves with natural dyes and silica–phosphate–M (M = Cu, Fe, or Al) nano pastes, and a combination of both methods. The resulting cocoon filaments were characterized by their siliconization of filament fibers by using FTIR, XRD, and SEM-EDS methods. The yarn tensile strength, color quality, color fastness properties affected by the siliconization of silk filament fibers, and antibacterial properties were also investigated. Results showed that the combination method produced better siliconization of silk fibers, and, consequently, the better siliconization of silk fibers produced better natural dyeing as well as antibacterial properties of their resulting silk yarns.

## 1. Introduction

Currently, natural silk fiber is attracting the world’s attention as a candidate for multifunctional intelligent biomaterial. On the other hand, natural silk threads and fabrics with natural dyeing are increasingly popular in the fashion industry, especially the environmentally friendly production process as well as their development into multifunctional smart biomaterials—for instance, in fields of pharmaceuticals and health and electronics and optical fibers. The research progress that has been reported in the last couple of years related to silk fiber as a candidate for multifunctional smart biomaterial in addition to being a raw material for clothing and fashion includes the dyeing of silk yarn with various natural colors assisted by fixators or mordants [1]; binding of Ag [2], Au [3], and Cu nanoparticles [4] to make them smart biomaterials; siliconization of silk fibers [5,6] to strengthen silk fibers; and cultivation of silkworms by feeding them mulberry leaves with added dyes [7,8] and minerals [9] in order to produce cocoons whose silk fibers are naturally colored. However, the current results of those studies revealed a need to develop an appropriate method for improving silk yarn quality and natural coloring and covering green industry and sustainability issues.

The advancement of silk fiber dyeing technology cannot be separated from dyeing technology. Dyeing is the process of applying color to textile materials evenly using a water medium. The dyeing process mechanism [10] includes (a) migration, which is the process of dissolving dyes in water solvents so that the dyes spread homogeneously to all parts of the solvent; (b) adsorption, which is the process of absorption of dyes by fibers and wherein the dye molecules have enough energy to resist the repulsive force of the chemical groups on the surface of the fibers; (c) diffusion, which is the entry of dyes from the surface of the fibers into the fibers; and (d) fixation, which is the binding of the auxo-chrome groups of dyes with the binding groups of fibers. The dyeing of silk yarn can be performed (a) without the use of a fixator, where the silk is directly dyed in a color solution; (b) through prefixation, where the silk fabric is soaked in a fixator solution first and then dipped in a color solution; (c) through post-fixation, which is the process of dyeing silk first into a color solution, then placing it into a fixator or mordants solution; and (d) through simultaneous fixation, where the color solution is mixed with the fixator solution and then the silk is dipped into the mixture so that the process of color absorption and fixation occurs simultaneously [11].

Some of the previous inventions as prior arts related to the natural dyeing process of natural fibers are the following patents: US007226647B2 [12], USOO8697429B2 [13], and US009506187B2 [14]. The results of the patent search obtained a summary of prior arts containing the stages of the preparation process or pretreatment of textile fibers to be dyed, the manufacture of natural color solutions, the mordanting process with varying stages, the dyeing process with variations in temperature and concentration of color solutions, and the accompanying process related to the fixation process using temperature and chemical compounds such as acids, bases, and salts. The dyeing process that became the invention still has weaknesses, namely, (1) the lack of effectiveness in absorbing natural colors and mordants so that it leaves a lot of dyeing waste and is less environmentally friendly; and (2) the quality of the tensile strength and creep of the yarn tends to decrease due to the dyeing and mordanting process so that it interferes with the quality of the process of using yarn or fabric afterwards. Based on previous findings, the invention to be carried out is to find a process of simultaneous natural coloring and fixation (using a natural color paste mixed with silica–Cu fixator nano paste) during the processing of cocoons into fiber and spinning and reeling the fiber into yarn. In addition, the process of dyeing and fixation and strengthening of fibers is carried out using the cultivation of silkworms with the addition of mulberry leaf feed with natural color paste, fixators nano paste silica–Cu, and the biomineral Ca–phosphate from bovine bone waste hydroxyapatite, and or a combination of the two.

The aim of this research was to apply nanomaterials in order to produce naturally colored natural silk yarn having antimicrobial properties and siliconized silk fibers for prospective multifunctional biomaterials as well as to develop eco-friendly production technology as a manifestation of the green industry movement. The novelties of this work are naturally colored and antimicrobial silk fibers and the production process is environmentally friendly through the direct dyeing process using natural dyes and silica–Cu fixator nano paste “in situ” during the processing of cocoons into silk yarn and from the cultivation of silkworms to produce cocoons through feed innovation (mulberry leaves enriched with natural dyes, silica–phosphate–Cu nano paste as a fixator as well as an additive). The two environmentally friendly production processes and/or a combination of them have inventive novelties and commercial feasibility, namely, making the dyeing process effective, reducing dyeing waste, and using local raw materials such as silica ash of rice husks, calcium phosphate from cow bone waste, and copper sulfate as a source of Cu.

## 2. Materials and Methods

The research design adopted a 4-step model: concept–product development–alpha and beta testing–launching according to Blank [15]; and implements function-oriented model-based product development according to Jacobs et al. [16]. The experimental series involving silkworms of *Bombyx mori* were already given ethical approval by Brawijaya University with the corresponding ethical approval code 133-KEP-UB-2024. The research was implemented in the following stages: (1) Preparing a production process design (production technology prototype design A or PTPD-A (Figure 1) and production technology prototype design B or PTPD-B) (Figure 2) natural silk yarn with natural color and antibacterial properties; (2) Making natural color fixator nano pastes based on nano silica from rice husk ash and copper sulphate and calcium phosphate of cow bone waste with various compositions; (3) Experimenting on the application of PTPD-A, namely, dyeing with mango leaf green natural color paste with various silica-Cu-phosphate fixator nano paste compositions during cocoon processing and yarn spinning at warm water temperature (40–60 °C) with the same repeated cocoon dyeing time for 1 h; (4) Experimenting on the application of PTPD-B, namely, silkworm cultivation with mulberry leaf feed that has been dipped in a mixture of natural color paste, nano silica-Cu paste and calcium phosphate additives from hydroxyapatite of cow bone waste, and then the resulting cocoon is processed into natural silk yarn by the same processing method; (5) Characterizing the nano pastes and natural dye pastes and silk yarn products resulting from the application of PTPD-A and PTPD-B with their respective appropriate instruments such as FTIR, XRD, XRF and SEM; (6) Testing of color quality (color difference and color aging), color fastness after washing, and sunlight exposure, as well as testing the tensile strength and creep strength of silk yarn resulting from PTPD-A and PTPD-B; (7) Testing of antibacterial properties (Gram-positive and Gram-negative) of PTPD-A and PTPD-B silk yarns.

An explanation of the PTPD-A chart in Figure 1 is as follows.

K = selected cocoon cultivated silkworms that were not damaged, and not double. 

A = the process of boiling cocoons at temperature and withdrawing silk fiber filaments from cocoons by the following steps, namely, (1) placing the selected cocoons into a boiling vessel that had previously been filled with warm water of 60–75 °C with a ratio of the number of cocoons and boiling water 1:1, or all cocoons could be fully submerged, then boil cocoons at 80–100 °C for 10–15 min until the cocoons appeared clean and the boiling water became cloudy, and (2) pulling the silk fiber filament from the cocoon and performing reeling, i.e., pulling the tip of the cocoon of as many as 16 or 32 cocoons and fusing each tip with a reeling and twisting machine. 

B = Reeling and twisting process of 16 or 32 silk fiber filaments into single yarns or Bs. 

C = Degumming process to remove gum or wax and sericin thin layer that was still attached to the surface of the thread (degumming) to the single thread by soaking the single thread (Bs) in a warm solution of soda ash or sodium carbonate 0.1–0.5% *m*/*v* (temperature 60–80 °C) for 15–30 min until a clean shiny single thread is obtained.

D = Natural dyeing process of dyeing single yarn with natural color nano paste (with the ratio of single yarn mass and natural color nano paste mass and the volume of water being 1:1:1–2) by soaking for 5–6 h.

E = The fixation process of soaking the colored double yarn in a mixture of silica–phosphate–mordant natural color fixator nano paste and natural color pairs used in dyeing (with a ratio of the mass of the fixator nano paste and the volume of water being 1:1–2) with a soaking time of 1–2 h in the fixation vessel. If the desired double yarn color intensity had not been obtained by adding nano paste natural color and natural color fixator nano paste on each vessel and dyeing per step D and fixating per step E, these two processes were repeated until the desired color intensity was obtained.

F = The process of rinsing the dyed and fixated yarn with clean water repeatedly until a clear rinse water was obtained.

G = The process of winding and drying the yarns resulting from the natural dyeing process by hanging the yarns rolled in a shady place and letting them dry.

The process of cultivating silkworms with feed engineering (Figure 2) was conducted in the following stages:The stage of hatching the eggs of the silkworm *Bombyx mori* (P0) was conducted by placing the silkworm in a place not exposed to sunlight for 10–12 days, then placing the silkworm seeds in a dark atmosphere with a humidity of 80–90% and an air temperature of 25–28 °C for 2 days, and preparing a sterile hatchery through the addition of chlorine powder (CaOCl_2_) and calcined lime (CaO) with a mass ratio of 5–7:95–93 that was evenly sprinkled over the surface of the newly hatched silkworm cluster.The stage of preparing natural color nano paste (W0) was conducted by extracting natural colors from natural materials, such as yellow natural color extract from turmeric rhizomes (*Curcuma longa* Linn.), red color extract from areca nut fruit (*Areca cathecu*), and blue color extract from indigo leaves (*Strobilanthes cusia*), then thickening the extract by heating slowly at low to medium temperature and/or adding coagulants such as CaO solution for indigo leaf extract until a natural color paste is obtained. Then, we prepared nano silica-based natural color fixator nano paste (F0) by mixing nano silica (made from rice husk ash) with nano calcium phosphate (made from hydroxyapatite of cow bone waste) and copper sulphate mineral with a ratio of 500–600 g of nano silica powder: 200–300 g of nano calcium phosphate powder: 100–200 g of copper sulphate (CuSO_4_), then made a mixture of natural color nano paste and natural color fixator nano paste (WF0) with a mass ratio of 1:1, using this mixture as an additional ingredient stock (WF0) in silkworm feed engineering.The stage of maintaining silkworms in the 1st life phase of instar 1 (I1) began the 1st day after hatching and lasted until the 4th or 5th day, when the small caterpillars fell asleep. It included preparing young mulberry (*Morus alba*) leaves, namely, the third to fifth leaves of the mulberry shoots, by washing the mulberry leaves with water, air-drying the washed mulberry leaves by placing the leaves in the shade until dry, and cutting or slicing the dried leaves to a width of 1–2 cm. Then, we fed the silkworms 3 times a day (morning at 07.00–08.00, afternoon at 12.00–13.00, night at 19.00–20.00), and repeating daily until the I1 ends, namely, the caterpillars begin to sleep, thus ending the I1 phase and progressing to the I2 (24–36 h) phase by evenly sprinkling the CaO mixed with CaOCl_2_ powders (95:5) on the surface of the caterpillars, while maintaining the room temperature and humidity during the 1st instar life phase of the caterpillars at 25–28 °C and 80–90%, respectively, by using air conditioning (air conditioning and air cooler) and spraying the walls of the room with water.The stage of raising silkworms in the 2nd life phase of instar 2 (I2) began with feed engineering through preparing fresh mulberry leaves from the 3rd to 10th leaf from the shoots, washing the leaves, then cutting or slicing them to a width of 3–5 cm. Then, we evenly sprayed a mixture of natural color nano paste and nano silica-based natural color fixator nano paste with a concentration of 1–1.25% *m*/*v* (or 10–12.5 mL of a mixture of natural color paste and nano silica-based natural color fixator nano paste in water until the volume becomes 1000 mL) (A1) on the surface of the sliced mulberry leaves, then we let the leaves sit in an open and shady space for 8–10 h until they are dry, and fed the silkworms 3 times a day (morning 07.00–08.00, noon 12.00–13.00, night 19.00–20.00). This is repeated daily until the I2 phase end, namely, the caterpillars begin to sleep. The 2nd instar silkworm progress to the I3 (24–36 h) phase by evenly sprinkling the CaO powder mixed with the CaOCl_2_ powder (95:5), while maintaining the room temperature and humidity during the I2 phase of the caterpillar life at 25–28 °C and 80–90%, respectively, by using air conditioning (air conditioner and air cooler) and spraying the walls of the room with water.The stage of raising silkworms in the 3rd life phase of instar 3 (I3) began with feed engineering through preparing fresh mulberry leaves from the 3rd to 10th leaf from the shoots, washing the leaves, evenly spraying a mixture of natural color nano paste and nano silica-based natural color fixator nano paste with a concentration of 2.5–3.0% volume/volume (or 25–30 mL of a mixture of natural color paste and nano silica-based natural color fixator nano paste in water until the volume becomes 1000 mL) (A2) on the surface of the mulberry leaf. Then, we let the leaves sit in an open and shady space for 8–10 h until they are dry, and feed the silkworms 3 times a day (morning 07.00–08.00, afternoon 12.00–13.00, night 19.00–20.00), repeating daily until the I3 phase ends, namely, the caterpillars begin to sleep, and the I3 phase progresses to the I4 (24–36 h) phase by evenly sprinkling the CaO powders mixed with CaOCl_2_ powders (95:5) while maintaining the room temperature and humidity during the I3 phase of the caterpillar life at 25–28 °C and 80–90%, respectively, by using air conditioning (air conditioner and air cooler) and spraying the walls of the room with water.The stage of raising silkworms in the 4th life phase of instar 4 (I4) began with feed engineering through preparing fresh mulberry leaves and stems/stalks with a size of 20–30 cm from the shoots, washing the leaves, evenly spraying a mixture of natural color nano paste and nano silica-based natural color fixator nano paste, with a concentration of 5–6% *m*/*v* (or 50–60 mL of a mixture of natural color paste and nano silica-based natural color fixator nano paste in water until the volume becomes 1000 mL) (A3), on the surface of the mulberry leaves and stems. Then, we let the leaves sit in an open and shady space for 8–10 h until they become dry, and feed the silkworms 3 times a day (morning 07.00–08.00, afternoon 12.00–13.00, night 19.00–20.00), repeating daily until the I4 phase ends, namely, the caterpillars begin to sleep. The 4th instar phase silkworms progress to the 5th instar (24–36 h) phase by evenly sprinkling the calcined lime powder (CaO) mixed with the chlorine powder (95:5) while maintaining the room temperature and humidity during the 4th phase of life at 25–28 °C and 80–90%, respectively, by using air conditioning (air conditioner and air cooler) and spraying the walls of the room with water.The stage of raising silkworms in the 5th life phase of instar 5 (I5) began with feed engineering through preparing fresh mulberry leaves and stems/stalks with a size of 50–60 cm from the shoots, washing the leaves, evenly spraying a mixture of natural color nano paste and nano silica-based natural color fixator nano paste with a concentration of 7.5–10% volume/volume (or 75–100 mL of a mixture of natural color paste and nano silica-based natural color fixator nano paste in water until the volume becomes 1000 mL) (A4) on the surface of the mulberry leaves and stems. Then, we let the leaves sit in an open and shady space for 8–10 until they become dry, and feed the silkworms 4 times a day (morning 07.00–08.00, afternoon 12.00–13.00, afternoon 16.00–17.00 and night 19.00–20.00), repeating daily until the I5 phase ends (about 6–7 days) with signs that the caterpillars would be nostalgic, namely, the caterpillars begin to stop eating, their movement slows down, the color of the caterpillars become bright yellow, and they secrete a saliva of fine fibers from their mouths, spin, and wrap themselves with the fibers that come out of their mouths. During the I5 phase, the temperature and humidity of the room is maintained at 25–28 °C and 80–90%, respectively, by using air conditioning and spraying the walls of the room with water.The cocooning stage (P1) was conducted by moving the caterpillars that were ready to nest to seeding sites (well known as Seri frames) that have been disinfected with lime powder mixed with chlorine in a ratio of 90:10 beforehand. Furthermore, we can harvest the cocoon after the 4th or 5th day from the time the caterpillar first grows and produces a cocoon (K) as expected.

The preparation of PTPD-C or natural silk yarn production technology that is natural in color and antimicrobial is completed through the above process of engineering mulberry leaf feed when silkworm cultivation is carried out, and then continued with the process of natural dyeing and fixation when the yarn is spun from the cocoon by considering the results of PTPD-A and its application as well as the results of the preparation of PTPD-B. We pay attention to the suboptimal natural coloring of single yarn or Bs as the result of the application of PTPD-A or PTPD-C at the stage of degumming, natural coloring, and fixation using double yarn or Bd, which is the doubling of single yarn with a doubling and twisting machine, as presented in Figure 3. The processes of PTPD-A, PTPD-B and PTPD-C were reported completed as Appendix A, respectively in the Appendix A.

The experimental testing resulted in single and double yarns that were color-fast, had yarn color difference, and showed color saturation and tensile strength and creep strength in the yarns. Color fastness testing used the Gray Scale standard, which assesses the color change in the specimen snippet by determining the level of color difference or contrast from the lowest level to the highest level. The color fastness test in sunlight was carried out by cutting the thread and attaching it to cardboard paper in such a way that a plate measuring 10 × 20 cm or 5 × 10 cm was obtained, then the test thread plate was half covered with black paper so that sunlight could not hit the test object, and half of it was left exposed to sunlight. Color fastness to soap washing was tested using a soap solution (B29 bar soap) in distilled water with a concentration of 5 g/L of water. The measurement of color fastness was a scale where the color fastness value was 5 (very good), 4–5 (good), 4 (good), 3–4 (quite good), 3 (enough), 2–3 (poor), 2 (poor), 1–2 (bad), and 1 (ugly). We tested yarn color differences (L*, a*, b*, dE*ab) and yarn color saturation (reflectance = %R) using UV-PC spectrophotometer with UV-2401-PC specification, catalog number 206-82201-93 Shimadzu Corporation, Kyoto, Japan, and its SR-2200 instruction manual. Testing of tensile and creep strengths was carried out using TENSO LAB TEST tools with the specification MESDAN LAB S.p.a, Salo, Italy, tenso model 300 type; 168 E Series No. 397, where the chosen unit of tensile strength was gram (g) and creep strength was percentage (%).

## 3. Results

### 3.1. Implementation Test of PTPD-A and Its Product Characterization

The implementation test of PTPD-A was conducted by using silk cocoons produced from silkworm cultivation with feed engineering that added green paste of mango (*Mangifera indica* L.) leaf extract and silica–phosphate–CuSO_4_ nano paste, three types of natural color pastes, and their respective fixator nano paste partners. These were (1) natural yellow color nano paste made from turmeric rhizome extract (*Curcuma longa* L.) with its fixator partner of nano paste silica–phosphate–alum; (2) a natural red color nano paste from areca nut (*Areca catechu*) extract and its fixator partner of silica–phosphate–CuSO_4_ nano paste; and (3) natural blue color nano paste made from local indigo plant (*Strobilanthes cusia*) leaf extract and its partner fixator, namely, silica–phosphate–FeSO_4_ nano paste. The results of PTPD-A were single white yarns without natural dyeing (symbolized as Bs), single dyed yarns with dyeing of natural turmeric yellow nano paste (BsK), single dyed yarn with natural red nano paste of areca nut extract (BsM), and single dyed yarns with blue nano paste of indigo plant leaf extract (BsB), as depicted, respectively, in Figure 4, with their characterization results using FTIR presented in Figure 5, and their summarized data in Table 1.

Based on the data in Table 1, the peaks that arise from the FTIR spectra and their relationship with the vibration modes of the bonds of the functional groups that may exist in the sample show the presence of siliconization (binding of silicon -Si- or silica -Si-O- and O-Si-O elements on silk fibers) and phosphitylation, namely, the presence of P-O-P and P-O-H groups and the bonding of Cu ions to the silk fibers. Special things that identify the presence of oxide or hydroxide bonds from mordant compounds such as Al-O-H on turmeric yellow dyed threads are due to the use of the nano paste fixator of silica–phosphate–alum, while blue natural-colored threads using silica–phosphate–FeSO_4_ fixator nano paste show the presence of Fe-O-H bonds.

The results of the tensile strength (g) and creep strength (%) of single threads that are naturally colored and not naturally colored are presented in Table 2 and depicted in Figure 6, and the results of the color fastness test against soap washing and against sunlight are presented in Table 3.

Based on those data, the natural dyeing of single threads using silk yarns created by silkworms cultivation with the most optimal feed engineering showed that the tensile and creep strength of turmeric yellow (BsK) and areca nut red (BsM) dyed single threads were not significantly different from the control yarn, namely, white single yarn Bs, while indigo plant leaf blue extract colored single yarn (BsB) had a relatively lower tensile strength value than the control. The color fastnesses to soap and sunlight washings from the yellow, red, and blue single yarns test specimens have average values of good to very good, but the turmeric yellow dyed yarns and the natural blue indigo colored yarns are quite different from the red areca nut yarn, which is more resistant to sunlight, as shown in Table 3. Meanwhile, Table 4 presents data on the color difference and intensity values of the applied test results. The intensity of natural red colored yarns of BsM is much better than natural blue colored yarns of BsB as well as natural yellow colored yarns of BsK.

### 3.2. Implementation of PTPD-B Test and Product Characterization

The implementation of the PTPD-B test was carried out in stages consecutively as follows: the demonstration plot of mulberry plant cultivation; the preparation of cages and hatcheries for silkworm eggs; the preparation of natural color nano pastes and silica–phosphate–Cu, silica–phosphate–Fe, and silica–phosphate–Al nano pastes as additional materials for the feed engineering process, as shown in Figure 2; the egg hatching process; the 1st instar nurturing process; the 2nd instar nurturing process with A1 engineered feeding; the 3rd instar nurturing process with A2 engineered feeding; 4th instar nurturing process with A3 engineered feeding; 5th instar nurturing process with A4 engineered feeding; caterpillar weight and length measurement; cocoon harvesting process; and cocoon length and weight measurement randomly taken for as many as 30 cocoons for each treatment and control. Figure 7 and Figure 8 display data on the average weight and length of silkworms at the end of the 5th instar life phase and the average weight, length, and diameter of cocoons produced from the silkworm cultivation process with various feed engineering treatments, as well as their comparison with the results of control cultivation carried out without providing feed engineering. Based on the data generated by random sampling techniques for each treatment, it was determined that the weight (g) of the cocoon produced was better than the control, and an improvement in the length and diameter of the cocoon occurred due to the feed engineering treatment. The administration of areca nut red natural color nano paste produced the best results compared to other treatment groups and controls.

The measurement of the weight and length of the filaments each cocoon produced was carried out by drawing filaments from five cocoons taken randomly for each treatment and measuring the weight (g) and length (m) of the resulting silk fiber filaments, with the data shown in Table 5.

Table 4 shows that the experimental group using the addition of red natural color of areca nut extract produced the highest fiber length compared to the other experimental groups and also compared to the control group.

### 3.3. Implementation of PTPD-C Test and Product Characterization

The implementation of PTPD-C test (Figure 3), that was a combination of PTPD-B and PTPD-A, consecutively produced double yarns, encoded as Bd for the control group, BdK for the double yarns dyed by turmeric yellow paste, BdB for the double yarns dyed by indigo blue paste, and BdM for the double yarns dyed by red areca nut paste, as depicted in Figure 9.

Color fastnesses against soap washing and sunlight tests as well as color and intensity differences and tensile and creep strength tests resulted in the data depicted in Table 6 and Table 7 and Figure 10. In comparison to the data of single yarns produced by PTPD-A, as depicted in Figure 6 and Table 2, Table 3 and Table 4, the double yarns produced by implementing PTTD-C have much better values than those test results.

Based on the data in Figure 11 and Table 8, the peaks that emerge from the FTIR spectra and their association with the vibration modes of the bonding of the functional groups that may be present in the samples indicate the presence of siliconized bonds (bonding of silicon-Si- or silica-Si-O- and O-Si-O elements on silk fibers), phosphating (phosphate bonding of yarns with the appearance of peaks identified as P-O-P and P-O-H bonds), and Cu bonding with the samples.

SEM-EDS microscopy scanning showed the microstructure, both morphology and homogeneity, of the grains of natural dye nano pastes and fixator nano pastes attached to silk fibers, as shown in Figure 12, Figure 13, Figure 14 and Figure 15.

Based on SEM-EDS data, it has been proven that siliconization occurs in silk yarn cultivated by silkworms with feed engineering which is followed by natural dyeing using natural color nano paste and natural color fixator nano paste during yarn spinning from cocoon or double silk yarn. This is a silk yarn produced after the process of reeling filaments from cocoons followed by doubling and twisting 16 filaments into single yarns, and doubling and twisting single yarns into double yarn 32/2. Before natural dyeing, the yarns were degummed, and after dyeing, these 32/2 yarns were then doubled and twisted again into colored natural silk yarn type 64/2 as one of the raw yarns for weaving crafts. The SEM results also showed the adhesion of natural color particles or fixator particles on the surface of silk fibers with an even distribution and nanoscale particle size.

Testing of the bacterial inhibition of the yarn samples of the control group as well as BdK, BdM, and BDB experimental groups, respectively, was carried out using the agar diffusion method as reported elsewhere [24] using pathogenic bacteria *Staphylococcus aureus* (Gram-positive) and *Escherichia coli* (Gram-negative) and an antibiotic comparator, ampicillin (Figure 16). The data from the bacterial inhibition tests are presented in Figure 17.

Figure 17 shows that the silk yarn produced by PTPD-C with red dye of areca nut extract (BdM) has the greatest inhibitory ability against both Gram-positive and Gram-negative bacteria with a percentage of 65.9% and 77% of the inhibitory power of ampicillin antibiotic bacteria, respectively.

## 4. Discussion

Novel natural dyeing processes of silk fiber using nano biomaterials in forms of nano pastes of silica–phosphate–alum, silica–phosphate–copper (II) sulfate, and silica–phosphate–iron (II) sulfate with respect to fixators of “in situ” natural dying of cocoon silk filaments when they are reeled into single silk threads, with natural green dye paste of mango leaf extract, natural yellow dye paste of turmeric extract, and natural red dye paste of areca nut extract, were reported as a prototype technological process design (PTPD-A). PTPD-A resulted in better quality of natural dyeing on silk fiber, as shown in Table 2 and Table 3, as well as retaining the tensile and creep strengths (as shown in Figure 6) of those single yarns from damage caused by the natural dyeing processes. They might also cause the formation of bridging bonds among chromic functional groups of responsible dyeing compounds of those extracts and hydroxyl and/or -N-H functional groups of protein compounds of silk fibers, as confirmed by the presences of O-Si-O, Si-O-H, and Si-O peaks of FTIR as well as P-O-H and P-O-P peaks of FTIR spectra depicted in Figure 5 and Table 1 that refer to cross-linking bonds, known as siliconizing and phosphatizing, respectively. The presence of Si-O-H in the silk fibers can improve their water absorption and water contact angle on silk fiber surfaces [24] to better bind with molecules of natural dyes. Meanwhile, the silica cross-linking between silk fibers and natural dye molecules can improve the interactive strength among them; thus, they can overcome the color fading problems of dyed silk fabrics as well as protect the silk intermolecular bonds from breaking, caused by the dyeing processes, which leads to wrinkles in silk fabrics [25]. The crosslinking agents such as nano silica and nano phosphate (in this study, the silica–phosphate-based fixator nano pastes) might take on an important role in maintaining the links between the chains in order to increase wrinkle resistance [25] as well as maintaining the tensile and creep strength of natural dyed silk yarn. The presence of Cu-O, Al-O-H, and Fe-O-H bonds on natural dyed silk fibers, as shown by the FTIR measurement data depicted in Table 1 and Figure 5, may also be responsible for better color fastness against soap washing and sunlight. It is likely related to the study of the effect of metal oxides on silk fiber resistance toward chemicals and UV rays of sunlight [26].

Nano materials application on silkworm *Bombyx mori* feed engineering using fresh mulberry leaves as a prototype technological process design is called PTPD-B here. The addition of natural dye pastes and silica–phosphate–M (M = Cu, Al, and Fe, where Cu is from CuSO_4_, Al from alum, and Fe from FeSO_4_) nano pastes sprayed on the surfaces of fresh mulberry leaves before silkworm feeding maintained healthy silkworms, as shown by the data of weight and length of *Bombyx mori* caterpillars, as depicted in Figure 7, and produced better cocoon and silk filaments, as depicted in Figure 8 and Table 4. The natural dyes used in these experiments, namely, natural green paste from mango leaf extract, yellow paste from turmeric extract, and red paste from areca nut extract, contain some bioactive compounds that are responsible for antibacterial and antifungal properties in protecting the cultivation of silkworms. The main component of mango leaf extract, mangiferin, for instance, is commonly used for naturally coloring Batik crafts in Indonesia because of its natural green color as well as its pharmacological benefits in protecting fabrics from free radicals and microorganisms [27], such as bacteria and fungi. Moreover, areca nut extract contains mainly tannins that can perform antibacterial activity because they interact with the bacterial cell wall to disrupt its integrity or regulate the bacterial metabolism [28]. Curcumin, the bioactive compound extracted from turmeric plants, is also a superior antibacterial agent [29]. The addition of nano silica, nano calcium phosphate, and CuSO_4_ as well as FeSO_4_-alum on the surface of mulberry leaves may support some silkworm metabolisms to become more productive in producing silk filaments with their cocoons. However, the exact rules and their mechanisms should be further investigated.

The implementation of PTPD-C, a combination of PTPD-B and PTPD-A, produced natural dyed double twisted silk yarns that have better quality color and improved fastness against soap washing and sunlight compared with the natural dyed single twisted yarns created by the application of PTPD-A. Additionally, the silk yarns produced by PTPD-C demonstrated antibacterial function. The invention of the PTPD-C demonstrates some beneficial values of nanomaterials in the development of sericulture through supporting the effectiveness of natural silk production and natural dyeing of the produced silk fibers, especially of those nanomaterials used as cross-linking agents for not only enhancing the quality of natural silk dyeing, but also the new prospect of using silk fibers as multifunctional biomaterials. Consequently, further intensive research and development as well as commercial needs assessment should be conducted in order to advance nano-material-based sericulture technology for green and sustainable organic–inorganic hybrid nano biomaterials.

There are three main results of the process production technology of the aforementioned PTPD-A, PTPD-B, and PTPD-C, namely, their color fastness, mechanical, and antimicrobial properties of the modified silk fibers. The color fastness values against soap washing of the modified silk fibers are relatively consistent in quality with the nonmodified silk fibers, but with respect to the color fastness against sunlight, the modified silk fibers are much better than those of nonmodified silk fibers, as shown in Table 2 and Table 5. Furthermore, the siliconization and phosphorylation of silk fibers, as aforementioned in the FTIR analysis results (Figure 5 and Table 1 as well as Figure 11 and Table 6), affect the mechanical properties, namely, tensile and creep strengths of modified silk fibers, as shown in Figure 6 and Figure 10. From the data in Figure 6, the modified single yarns (produced by PTPD-A) with natural dyed red extracted from areca nut (sample code of BsM, red colored single yarn) have better mechanical properties than those of the controlled single yarn sample (code Bs, without natural dyeing when the filament was reeled from cocoon), and much better than those of BsK (natural yellow dyed when the filament was reeled from cocoon) and BsB (natural blue dyed when the filament was reeled from cocoon). The cocoons used in these dyeing processes were natural cocoons produced from normally treated silkworm cultivation. Meanwhile, the cocoons that resulted from feed engineering process of PTPD-B and continued with “in situ” natural dyeing process directly when the cocoons were reeled (PTPD-A) produced double yarns, called PTPD-C (code samples of Bd, BdK, BdM, and BdB for using natural dyeing, natural yellow, red and blue dyeing, respectively), after reeling and doubling processes. The tensile as well as creep strengths of double yarns are shown in Figure 10, and show the values of mechanical properties of BdM ≈ BdK > BdB > Bd. A comparison of the data in Figure 6 and Figure 10 shows that feed engineering and natural dyeing affect the mechanical properties of silk yarns. The modified silk fibers of BdK, BdM, BdB, and Bd samples of double yarns produced from the implementation of PTPD-C (a combination of PTPD-B and PTPD-A) have good antimicrobial properties against Gram-positive bacteria of Streptococcus aureus and Gram-negative bacteria of Escherichia coli, but the antibacterial properties were not as strong as the positive control by using the antibiotic of Ampicillin. The aforementioned data comparison inferred that the natural-colored silk fabrics composed of the modified silk fibers produce better longevity and luxurious characteristics because of better color fastness, mechanical, and antimicrobial properties.

The modified silk fibers, namely, natural dyed and siliconized, phosphorylated, Cu-enriched silk filaments, produced in the PTPD-C are proposed to have better life cycle environmental impacts because of organic farming of silkworm, significantly reducing environmentally toxic chemicals throughout the modified silk yarn production, and the use of nanomaterials enhancing the effectiveness as well as efficiency of the whole processes. The modified silk yarn production has a business life cycle that consists of mulberry plantation, feed engineering preparation, silkworm hatchery from instar 1 to instar 5, cocooning process, cocoon harvest and treatment, silk filament reeling with “in situ” natural dyeing of the filament into single yarn, doubling the yarn with direct natural dyeing to become natural dyed double yarns, and weaving process with natural dyed silk fabrics. In the mulberry plantation, the organic farming uses organic fertilizer and organic pesticides to improve the negative environmental effects. The use of nano pastes of silica–phosphate–Cu, prepared by using environmentally safe raw materials, such as nano silica prepared from rice husk ashes, nano calcium phosphate from hydroxy apatite of bovine bone wastes, and nanoparticle Cu produced by green synthesis using CuSO_4_ powder and mango leaf extract, was proposed to have better environmental impacts. The use of natural dye pastes, namely, turmeric rhizome extract for natural yellow dye paste, areca nut extract for natural red dye paste, and Indigofera leaf extract for natural blue dye paste, can also improve the environmental impacts because of the biodegradability of those dye pastes as well as minimal use of chemicals: only ethanol as well as lime solution for extracting processes and making dye pastes. Inorganic silica–phosphate–Cu, silica–phosphate–Fe, and silica–phosphate–Al fixator nano pastes used in the process of feed engineering as well as in the direct dyeing processes were prepared greenly by using nano silica powder prepared from rice husk ashes, nano calcium phosphate from bovine bone wastes, cupric sulphate, ferro-sulphate mineral, and alum minerals, respectively. The mulberry leaves were sprayed on both leaf surface sides with natural dye pastes, silica–phosphate–Cu nano paste, and inorganic natural dye fixators (silica–phosphate–Cu for natural yellow dye paste, silica–phosphate–Fe for natural blue dye paste and silica–phosphate–Al for natural red dye paste, respectively) when the feed engineering was started, to reduce the excess of those pastes used. Thus, it can produce better environmental impacts. In the silkworm cultivation processes from instar 1 to instar 5, the engineered food of fresh mulberry leaves was used from instar 4 until instar 5 with consideration for the matured caterpillars, while the silkworms from the beginning of instar 1 until the end of instar 3 were fed normally with fresh mulberry leaves. Thus, the process of feed engineering in silkworm cultivation is already considered animal-friendly treatment. The dry mulberry leaves and stems as well as silkworm moles in the bed of the silkworm hatchery were collected and fermented aerobically to become an organic fertilizer for better environmental impact. In the cocooning process, the matured and ready-to-cocoon caterpillars were placed in woven bamboo cocoon tools to reduce the use of plastic Seri frames, and in the process of filament reeling from cocoon, where a degumming process should be conducted, a soda ash solution in dilute concentration was used for degumming the silk fibers. The process was also eco-friendly and improved the environmental impact. However, the total energy and water used for the whole processes as well as the CO_2_ emissions has not yet been assessed and calculated accurately. Those environmental impact data should be compared with the life cycle assessment environmental impact data of other silk fiber productions as well as other fabric fiber productions [30], for instance, the assessment data on the environmental impact of silk production resulting in GHG emissions of 52.5 kg CO_2_ eq/kg and energy usage of 1467.3 MJ/kg [31] and water usage of 26,700 L/kg [32].

The invention of the feed engineering process and the product of modified silk fibers have potential for further investigations leading to prospective smart organic–inorganic hybrid nano biomaterials in areas of biomedical and pharmaceutical, bioengineering, and electronic and magnetic biomaterials. The modified silk fibers produced through feed engineering by modifying only mulberry leaves for feeding Bombyx mori silkworms as resulted in this study can be advanced through other experiments, for instance, using pellet feed forms comprising mulberry leaves and stems, other high-protein plants such as moringa leaves, and Indigofera leaves. In addition, soon, this could include additive bioabsorbable nano substances for antibacterial, antifungal, immunomodulator, growth enhancing, and bioattractant compounds for attracting the silkworm in feeding pellets and hydrostable compounds for the pellets to attain enough water to keep silkworms in good health. The modified silk fibers can also be used in prospective research on multifunctional organic–inorganic hybrid nano biomaterials in the area of biomedicals and pharmaceuticals; for instance, modified sericin, as part of modified silk fibers, has valuable prospects in future research because sericin can produce hydrogels, films, sponges, foams, dressings, particles, fibers, etc., for various biomedical and pharmaceutical applications (e.g., tissue engineering, wound healing, drug delivery, cosmetics) [33]. Moreover, the modified fibroin, the main part of the modified silk fibers, is becoming increasingly attractive to investigate currently as well as in the future; for instance, silk fibroin has potential use in the area of biomedicine as a bioactive agent in wound healing and skin regeneration [34]. Silk fibroin could also be investigated for its potential benefits as a drug delivery agent [35] and cancer therapy [36]. The modified silk fibers created by templating nanoparticles of metals as well as metal oxides can become futuristic electronic biomaterials in wider applications, as intensively reviewed by Ahmed et al. [37], as well as magnetic biomaterials, as reported currently by Eivazzadeh-Keihan et al. [38]. The remarkable future of modified silk fibers lies in research in biomedicine, pharmaceuticals, bioengineering, and electronic and magnetic biomaterials based on modified silk fibers in the form of organic–inorganic hybrids.

## 5. Conclusions

The combination method (called PTPD-C here) of silkworm feed engineering through adding nano pastes of silica–phosphate–M (M = Cu, Al, or Fe) and pastes of natural dyes on the surfaces of fresh mulberry leaves (PTPD-B) and “in situ” natural dyeing processes when filaments of silkworm cocoons were reeled and twisted to become silk yarns (PTPD-A) shows the importance of nanomaterials in supporting the organic–inorganic hybridization among silk fibers, inorganic fixators, and natural dyes, leading to the production of superior natural dyed silk yarn quality, not only for supporting the industry of silk fabrics and fashions, but also in producing beneficial biomaterials. Moreover, the PTPD-C uses nano silica from rice husk ashes, nano calcium phosphate isolated from bovine bone wastes, and bioresources of natural dyes that can add value to local raw materials as well as promote green industry in producing prospective multifunctional biomaterials.

## 6. Patents

The Indonesian Patent P00202307610 filed on 16 August 2023 resulted from the work reported in this manuscript.

## Figures and Tables

**Figure 1 nanomaterials-14-01697-f001:**
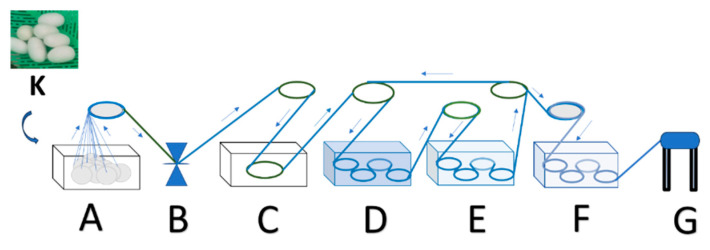
PTPD-A, where the blue line, wider blue circles, and green circles symbolize the processes involved in reeling process; smaller blue circles in the boxes symbolize that the threads are dyed and fixed with mordants.

**Figure 2 nanomaterials-14-01697-f002:**
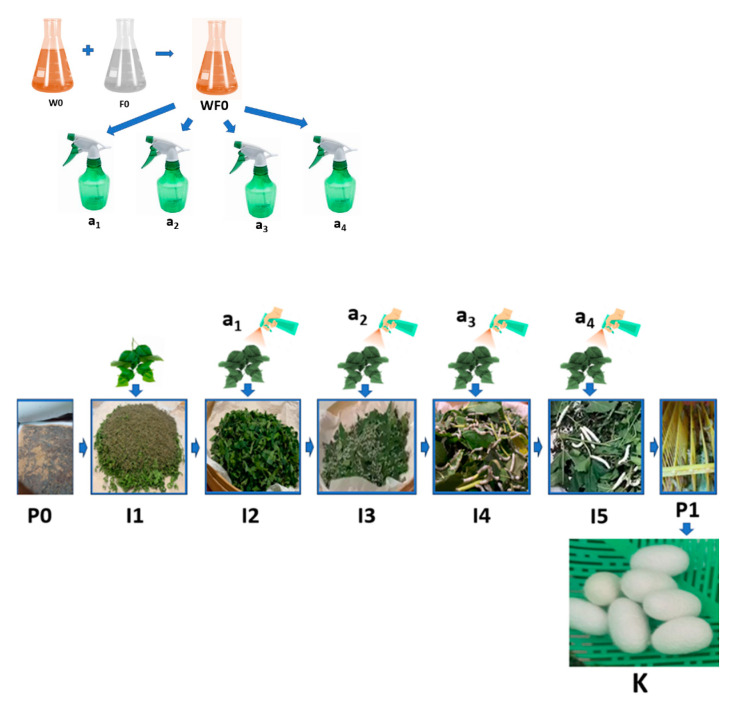
PTPD-B.

**Figure 3 nanomaterials-14-01697-f003:**
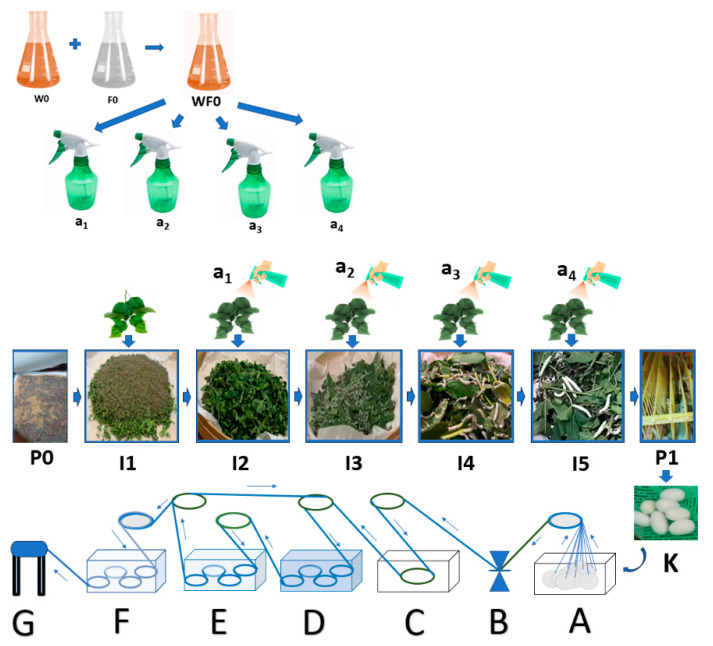
PTPD-C.

**Figure 4 nanomaterials-14-01697-f004:**
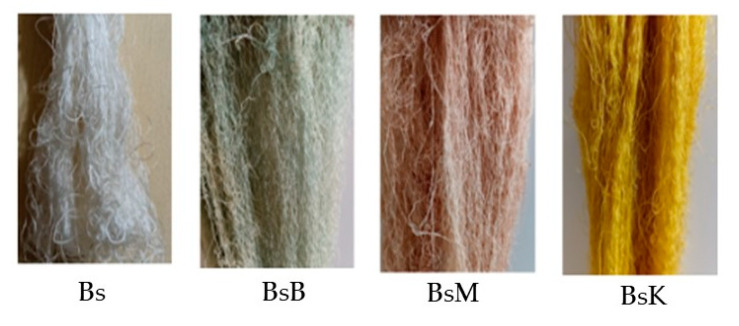
Natural dyed single yarns products of PTPD-A.

**Figure 5 nanomaterials-14-01697-f005:**
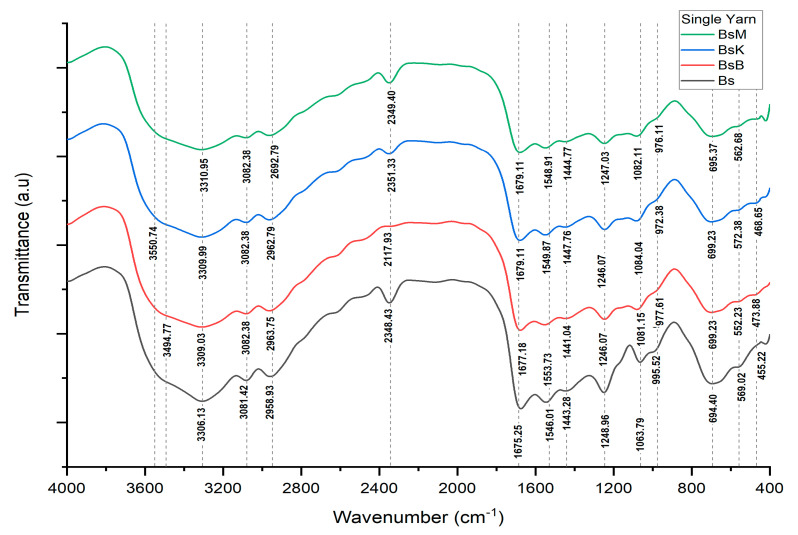
FTIR spectra of PTPD-A products.

**Figure 6 nanomaterials-14-01697-f006:**
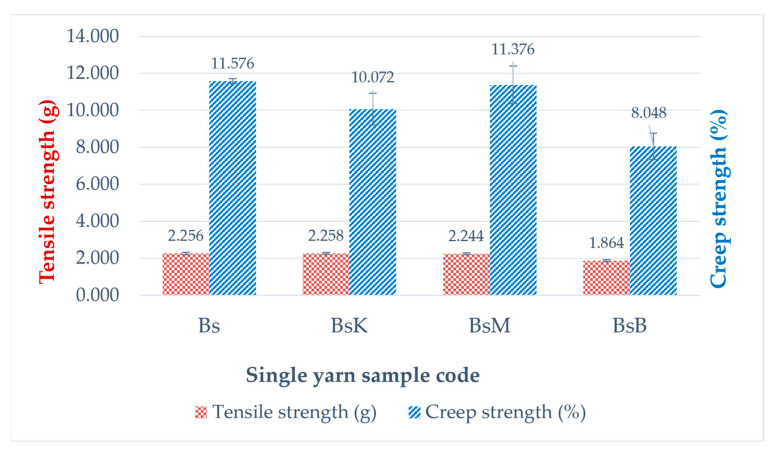
Tensile and creep strength of single yarns.

**Figure 7 nanomaterials-14-01697-f007:**
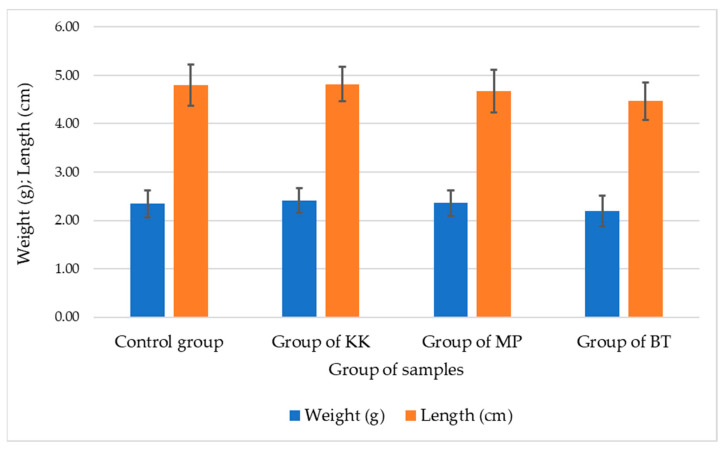
Graph of the average weight and average length of silkworms at the end of the 5th instar.

**Figure 8 nanomaterials-14-01697-f008:**
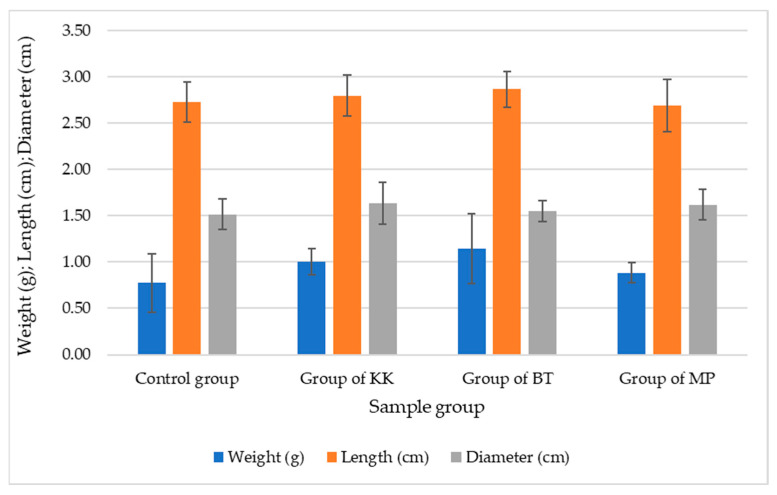
Comparison of the average weight, length, and diameter of the resulting cocoon.

**Figure 9 nanomaterials-14-01697-f009:**
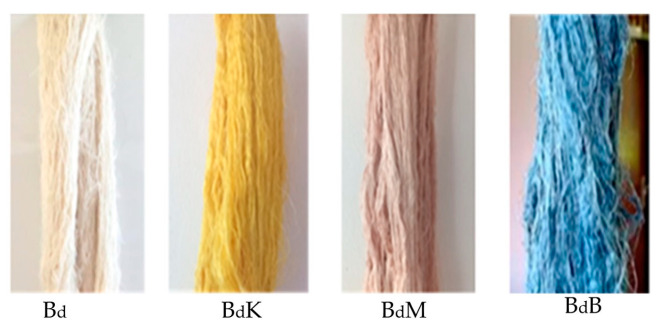
Natural dyed double yarn products of PTPD-C.

**Figure 10 nanomaterials-14-01697-f010:**
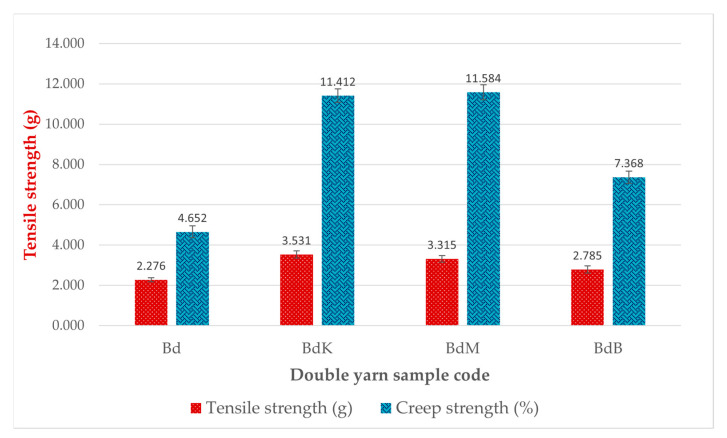
Tensile and creep strengths of double yarn samples.

**Figure 11 nanomaterials-14-01697-f011:**
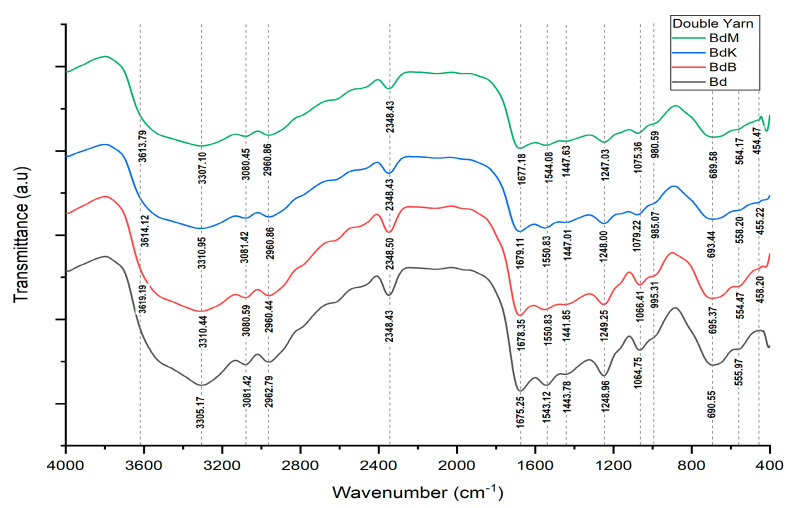
FTIR spectra from double yarn samples from PTPD-C applied test.

**Figure 12 nanomaterials-14-01697-f012:**
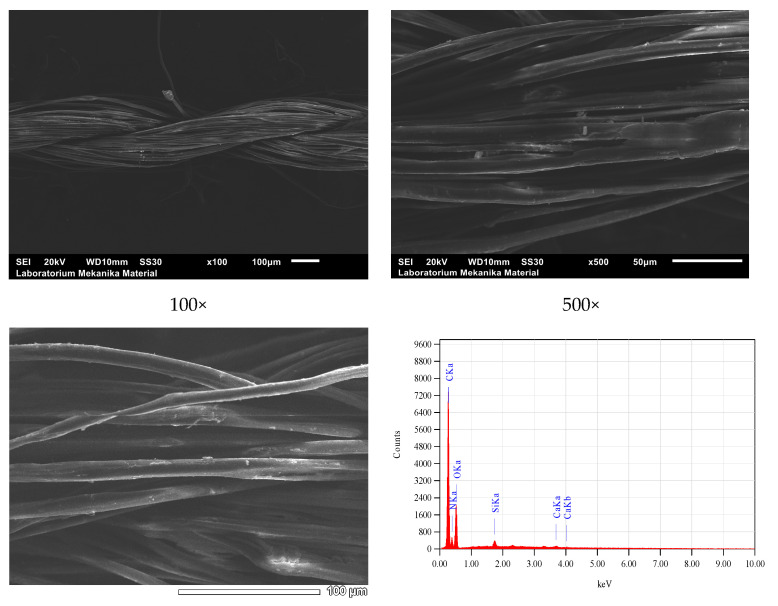
SEM micrograms and EDS spectra of Bd yarns, where red line of EDS spectra indicating the elemental distribution observed in the sample..

**Figure 13 nanomaterials-14-01697-f013:**
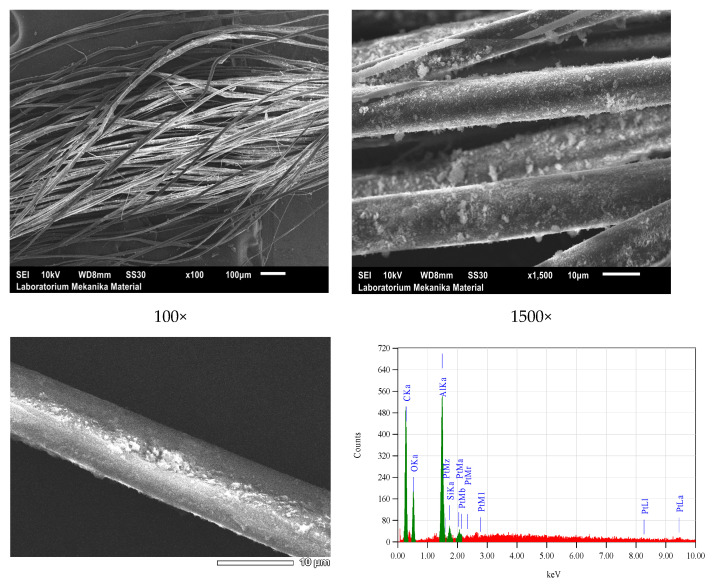
SEM micrograms and EDS spectra of BdK yarns.

**Figure 14 nanomaterials-14-01697-f014:**
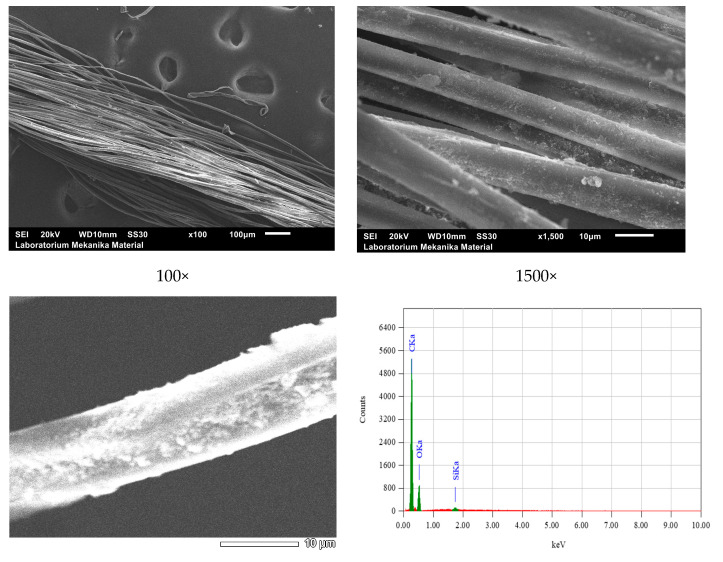
SEM micrograms and EDS spectra of BdM yarns, where red line of EDS spectra indicating the elemental distribution and green line the dominant elements observed in the sample.

**Figure 15 nanomaterials-14-01697-f015:**
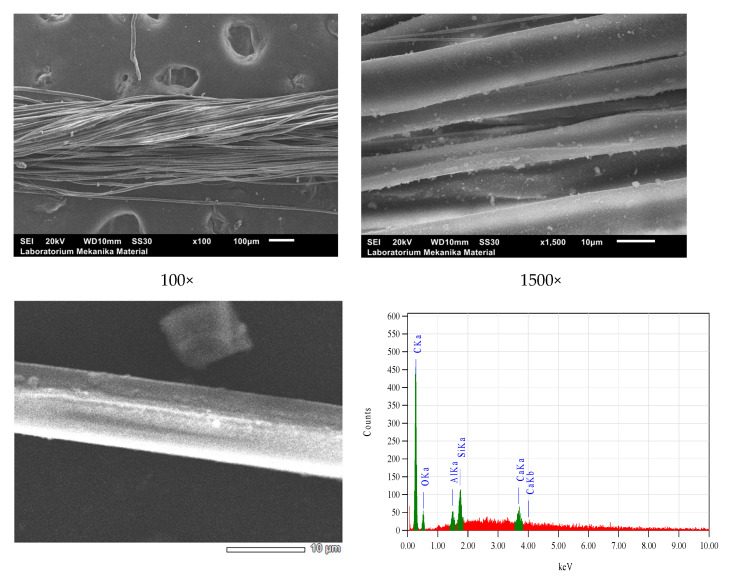
SEM micrograms and EDS spectra of BdB yarns.

**Figure 16 nanomaterials-14-01697-f016:**
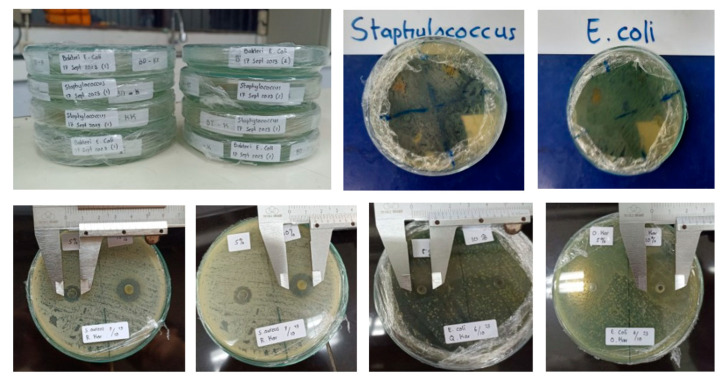
Testing of bacterial inhibition of silk yarn samples of PTPD-C results.

**Figure 17 nanomaterials-14-01697-f017:**
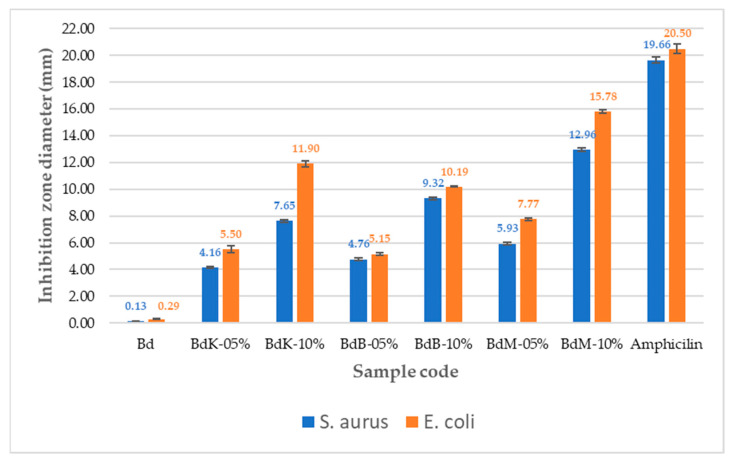
Inhibition zone diameter of double yarns resulted by PTPD-C.

**Table 1 nanomaterials-14-01697-t001:** Results of FTIR data analysis of yarn samples from PTPD-A applied test.

Bonding	Wave Numbers (cm^−1^)				Reference Wave Numbers (cm^−1^)
	Bs	BsK	BsM	BsB	
Cu-O	569.02	572.38	562.68	552.23	437–580 [17]
O-Si-O	455.22	468.65	−1082.11	473.88	467–532 bending [18] 1111–1114 asymmetric [19]
N-H	694.40	699.23	695.37	699.23	665–910 stretching [20]
P-O-H	995.52	972.38	976.11	977.61	910–1040 [20]
P-O-P	995.52	972.38	976.11	977.61	807–1000 [20]
C-N	1248.96	1246.07	1247.03	1246.07	1240–1340 [20]
N-O	1443.28	1447.76	1444.77	1441.04	1475–1550 asymmetric [19] 1290–1400 stretching [20]
N=N	1546.01	1549.87	1548.91	1553.73	1429–1576 [20]
O-H	1675.25	1679.11	1679.11	1677.18	1632–1742 [21]
Si-O-H	3306	3309.99	3310.95	3309.03	3200–3390 [21,22]
Si-O	1063.79	1084.04	1082.11	1081.15	1002–1034 [19]
Al-O-H	-	3550.74	-	-	3618–3696 [19]
Fe-O-H	-	-	-	3494.77	3425 [23]

**Table 2 nanomaterials-14-01697-t002:** Tensile and creep strengths of natural colored single yarn.

Sample Code	Tensile Strength (g)			Creep Strength (%)		
	Measured Data	Average	STDeV	Measured Data	Average	STDeV
Bs	2.308	2.256	0.1079	11.571	11.5760	0.1326
	2.328			11.711		
	2.132			11.446		
BsK	2.254	2.258	0.0656	11.155	10.0717	1.0548
	2.325			10.012		
	2.194			9.048		
BsM	2.211	2.244	0.0523	11.344	11.3763	1.0129
	2.304			10.380		
	2.216			12.405		
BsB	1.844	1.864	0.0510	8.143	8.0477	0.7098
	1.922			8.705		
	1.826			7.295		

**Table 3 nanomaterials-14-01697-t003:** Color fastness of yarns against soap washing and sunlight.

Sample Code	The Value of Color Fastness of Yarns Against Soap Washing (Gray Scale)	The Value of Color Fastness of Yarns Against Sunlight (Gray Scale)
Bs	-	-
BsK-1	4 (Good)	3–4 (Sufficient)
BsK-2	4 (Good)	3–4 (Sufficient)
BsK-3	4 (Good)	3–4 (Sufficient)
BsM-1	4 (Good)	4 (Good)
BsM-2	4 (Good)	4 (Good)
BsM-3	4 (Good)	4 (Good)
BsB-1	4–5 (Good)	3–4 (Sufficient)
BsB-2	4–5 (Good)	3–4 (Sufficient)
BsB-3	4–5 (Good)	3–4 (Sufficient)

**Table 4 nanomaterials-14-01697-t004:** Results of color difference and intensity testing.

Sample Code	Color Difference Test Value				Color Intensity Test Values (R%)
	L*	a*	b*	dE*ab	
Bs	91	1.08	−0.58	0.00	101.43
BsK-1	53.28	16.12	39.09	56.76	17.06
BsK-2	61.35	14.25	42.63	54.05	13.13
BsK-3	61.16	14.21	40.48	52.42	12.04
BsM-1	29.37	22.33	25.77	70.31	7.53
BsM-2	29.33	21.64	26.61	70.46	7.06
BsM-3	33.88	21.06	20.26	64	7.59
BsB-1	52.24	−2.06	−3.64	38.92	29.37
BsB-2	53.07	−1.1	1.1	38.26	27.07
BsB-3	53.28	−3.18	3.96	38.33	30.87

**Table 5 nanomaterials-14-01697-t005:** Weight and length of silk fiber filament of each cocoon.

Sample Number	Control Group		Experimental Group of KK		Experimental Group of BT		Experimental Group of MP	
	Weight (g)	Length (m)	Weight (g)	Length (m)	Weight (g)	Length (m)	Weight (g)	Length (m)
1	0.23	578.25	0.29	694.74	0.28	732.25	0.34	922.25
2	0.25	590.62	0.31	725.25	0.33	768.94	0.36	936.75
3	0.21	547.92	0.28	694.95	0.27	678.45	0.31	870.75
4	0.21	550.21	0.27	687.85	0.28	680.25	0.29	825.78
5	0.23	571.95	0.26	675.76	0.27	670.95	0.28	815.88
Average	0.23	567.79	0.28	695.71	0.29	706.17	0.32	874.28
St. Dev.	0.02	18.38	0.02	18.26	0.03	42.72	0.03	54.72

**Table 6 nanomaterials-14-01697-t006:** Natural dyed double yarns quality produced from PTTD-C implementation.

Sample Code	Test Nr.	Value of Color Fastness Against Soap Washing (Gray Scale)	Value of Color Fastness Against Sunlight (Gray Scale)	Value of Color Differences				Value of Color Intensity Test (R%)
				L*	a*	b*	dE*ab	
Bd		-	-	97.7	0.41	−2.49	0.00	193.3
BdK	1	4 (Good)	4 (Good)	77.88	35.75	40.31	60.79	34.2
	2	4 (Good)	4 (Good)	72.35	29.78	42.18	57.2	35.62
	3	4 (Good)	4 (Good)	72.22	35.15	36.42	58.04	39.08
BdM	1	4–5 (Good)	4–5 (Good)	54.69	30.89	−0.88	52.74	55.08
	2	4–5 (Good)	4–5 (Good)	55.51	27.93	5.64	51.03	59.79
	3	4–5 (Good)	4–5 (Good)	54.54	30.21	1.07	52.58	54
BdB	1	4 (Good)	4 (Good)	22.21	5.32	−28.24	80.01	38.18
	2	4 (Good)	4 (Good)	24.79	9.64	−34.56	80.19	30.72
	3	4 (Good)	4 (Good)	24.72	7.49	−34.46	80	39.9

**Table 7 nanomaterials-14-01697-t007:** Tensile and creep strengths of natural colored double yarn.

Sample Code	Tensile Strength (g)			Creep Strength (%)		
	Measured Data	Average	STDeV	Measured Data	Average	STDeV
Bd	2.387	2.276	0.1026	4.902	4.6517	0.2939
	2.185			4.725		
	2.255			4.328		
BdK	3.640	3.531	0.1729	11.062	11.4123	0.3344
	3.332			11.728		
	3.622			11.447		
BdM	3.215	3.315	0.1652	11.984	11.5840	0.3722
	3.506			11.248		
	3.225			11.52		
BdB	2.687	2.785	0.1738	7.068	7.3680	0.3056
	2.986			7.679		
	2.683			7.357		

**Table 8 nanomaterials-14-01697-t008:** Results of FTIR data analysis of double yarn samples from PTPD-C applied test.

Bonding	Wave Numbers (cm^−1^)				Reference Wave Numbers (cm^−1^)
	Bd	BdK	BdM	BdB	
Cu-O	555.97	558.20	564.17	554.47	437–580 [17]
O-Si-O	-	455.22	454.47	458.20	467–532 bending [18]
N-H	690.55	693.44	689.58	695.37	665–910 stretching [20]
P-O-H	-	985.07	980.59	995.31	910–1040 [20]
P-O-P	-	985.07	980.59	995.31	807–1000 [20]
C-N	1248.96	1248.00	1247.03	1249.25	1240–1340 [20]
N-O	1443.78	1447.01	1447.63	1441.85	1475–1550 asymmetric [19]
N=N	1543.12	1550.83	1544.08	1550.83	1429–1576 [20]
O-H	1675.25	1679.11	1677.18	1678.35	1632–1742 [21]
Si-O-H	3305.17	3310.95	3307.10	3310.44	3200–3390 [21,22]
Si-O	1064.75	1079.22	1075.36	1066.41	1002–1034 [19]
Al-O-H	-	3614.12	3613.79	3619.19	3618–3696 [19]
Fe-O-H	-	-	-	3310.44	3425 [23]

## Data Availability

All the data of this manuscript are included in the manuscript. No separate external data source is required. If anything is required from the manuscript, certainly, this will be extended by communicating with the corresponding author through corresponding official mail: karyasa@undiksha.ac.id.

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
