# Peer review of "Organic–Inorganic Hybridization of Silkworm Cocoon Filaments Using Nano Pastes of Silica–Phosphate–M (M = Cu, Fe, or Al)"

_nanomaterials, 2024, doi:10.3390/nano14211697_

Round 1
Reviewer 1 Report
Comments and Suggestions for Authors
This study aimed to explore methods for siliconization of silkworm cocoon filaments and assess their effects on yarn quality, natural dyeing, and antibacterial properties. Three siliconization methods were tested: in-situ natural dyeing during silk spinning, feed engineering with natural dyes, silica-phosphate-metal (Cu, Fe, or Al) nano-paste sprayed on mulberry leaves, and a combination of both. The degree of siliconization was characterized using FTIR, XRD, and SEM-EDS. The study also investigated the impact of siliconization on yarn tensile strength, color quality, colorfastness, and antibacterial properties. Results indicated that the combined method achieved superior siliconization, leading to enhanced natural dyeing and antibacterial performance of the silk yarns. In my opinion, it is recommended that this paper be published in the journal Nanomaterials after addressing the following issues.
1. I think many continuous citations in the literature are not rigorous, for example, in line 39, where a summative description uses seven literatures [1-7]. It is recommended that the literature be cited precisely where it is needed.
2. Determine if the corresponding ethics approval code is "XXX"? Please provide the approval document of the ethics review report.
3. In the infrared spectrogram of Figure 5 and Figure 10, the specific wavenumber marker numbers are fuzzy and illegible. It is recommended that they be modified to be legible. In addition, why not list the four spectral lines in a diagram? If the different spectral lines are compared in a diagram, the wavenumber changes of various materials should be more intuitive.
4. The wavenumber data in Table 1 should use the same number of decimal places.
5. The results of tensile strength (g) and creep strength (%) of a single or single line in Figure 6 should be averaged over several experiments. It should be stated clearly in the method, and the result should be the average plus the deviation.
6. The manuscript contains some inaccuracies in its presentation and illustrations. It is recommended that the full text be carefully examined and revised.
Comments on the Quality of English LanguageThe English language quality in the provided text is adequate but could benefit from further refinement.
Author Response
Comments 1: I think many continuous citations in the literature are not rigorous, for example, in line 39, where a summative description uses seven literatures [1-7]. It is recommended that the literature be cited precisely where it is needed.
Response 1: Thank you for pointing this out. We agree with this comment. Therefore, we have revised by choosing only one literature [1] from seven literatures [1-7] in line 39 as well as one literature [2] from three literatures [8-10] in line 39. Consequently, the next literature numbers as well as reference list are also changed.
Comments 2: Determine if the corresponding ethics approval code is "XXX"? Please provide the approval document of the ethics review report.
Response 2: We have, accordingly, added the ethic approval code 133-KEP-UB-2024 in line 102. We provide also the approval document on the link below:
https://drive.google.com/file/d/1meabeMxbvsCI_lMy-YGhjwuTjrL-9QED/view?usp=sharing
Comments 3: In the infrared spectrogram of Figure 5 and Figure 10, the specific wavenumber marker numbers are fuzzy and illegible. It is recommended that they be modified to be legible. In addition, why not list the four spectral lines in a diagram? If the different spectral lines are compared in a diagram, the wavenumber changes of various materials should be more intuitive.
Response 3: Agree. We have, accordingly, changed to emphasize this point by refining the Figure 5 and Figure 11 (changing number because of figure addition before it, see Response of comment 5), thus the specific wavenumbers are clearly read. We thank you for the suggestion, we agree to draw the four diagrams becoming one diagram as shown in Figure 5 in line 321 as well as Figure 11 in line 412.
Comments 4: The wavenumber data in Table 1 should use the same number of decimal places.
Response 4: Agree. We have, accordingly, changed to emphasize this point in Table 1 line 323 as well as Table 6 in line 418.
Comments 5: The results of tensile strength (g) and creep strength (%) of a single or single line in Figure 6 should be averaged over several experiments. It should be stated clearly in the method, and the result should be the average plus the deviation.
Response 5: Thank you for the suggestion. We have, actually already the three times data of tensile strength (g) as well as creep strength (%), thus we have changed the Figure 6 with the average and the deviation as shown in line 431. Furthermore, in line with your suggestion we add also the average and deviation of tensile strength (g) and creep strength (%) for double yarns as shown in Figure 10 in line 410 as a refinement of Table 5 in line 407. We stated also the method in line 282 to line 300.
Comments 6: The manuscript contains some inaccuracies in its presentation and illustrations. It is recommended that the full text be carefully examined and revised.
Response 6: Thank you for your comments and recommendation. We have, examined and revised our full text, with a hope that this revision of our manuscript matches and fulfill the quality standard of this journal.
Response to Comments on the Quality of English Language
The English language quality in the provided text is adequate but could benefit from further refinement.
Response: We thank you for the comments on the manuscript English quality. We have tried to revised and refined the whole text.

Reviewer 2 Report
Comments and Suggestions for Authors
This paper is like a scientific experiment report rather than a scientific research paper, and the English it too confused to be understand. Thus it not recommend to publish in nanomaterials.
1. Figures 1-3 are confusing and the caption reveals useless information.
2. The aim of this work can be more clearly introdunced in the introduction sections, as well as the novelty of this work.
3. The quality of Figure 5 is poor and hard to read.
4. The tensile strenght cures should provided in the main text. without the cures, Figure 6 is not trusted.
5. What's happened to the error bar in Figure 7? Some of them have strage back ground that pionting to potential counterfeiting (for instance, 0.77, 2.72, 1.52, 1.63, 2.86, 2.69, and 1.62)
6. The size of word in Figure 9 are too big.
Comments on the Quality of English Language
The english of this work is hard to read, and must be improved.
Author Response
Comments 1: Figures 1-3 are confusing and the caption reveals useless information.
Response 1: Thank you for pointing this out. In our gratitude for your comment, we do not agree with this comment. We try to explain in details our prototype of feed engineering models by using Figure1-3 and their consecutive explanation.
Comments 2: The aim of this work can be more clearly introduced in the introduction sections, as well as the novelty of this work.
Response 2: We have, accordingly, elaborated and revised this work to be more clear purposes and novelty as shown in yellow mark of text in line
Comments 3: The quality of Figure 5 is poor and hard to read.
Response 3: Thank you. We have, accordingly, changed to emphasize this point by refining the Figure 5 and Figure 11, thus the specific wavenumbers are clearly read. We thank you for the suggestion, we have already drawn the four diagrams becoming one diagram as shown in Figure 5 in line 321 as well as Figure 11 in line 412.
Comments 4: The tensile strength cures should be provided in the main text. without the cures, Figure 6 is not trusted.
Response 4: Thank you for the suggestion. We have, actually already the three times data of tensile strength (g) as well as creep strength (%), thus we have changed the Figure 6 with the average and the deviation as shown in line 431. Furthermore, in line with your suggestion we add also the average and deviation of tensile strength (g) and creep strength (%) for double yarns as shown in Figure 10 in line 410 as a refinement of Table 5 in line 407. We have already stated the method of measuring tensile and creep strengths in line 297 to line 300.
Comments 5: What's happened to the error bar in Figure 7? Some of them have strange back ground that pionting to potential counterfeiting (for instance, 0.77, 2.72, 1.52, 1.63, 2.86, 2.69, and 1.62)
Response 5: Thank you, we have already revised the graph by removing the pointing back ground. We would like to explain the diagram 7 dan 8. The number on upper the bars are the data labels not the error bar labels. For instance, on Figure 8, in control group, weight of silkworm (g) = 0.77±0.32; length (cm) =2.72±0.22, and diameter of silkworm (cm) = 1.52±0.16 and soon. These data fulfilled statistically measuring 30 silkworms randomly sampled from about 100-150 silkworms for each treatment. Our original data of measurement is below.
Comments 6: The size of word in Figure 9 are too big.
Response 6: Thank you for your comments. We have, accordingly, revised the word in Figure 9 as well as Figure 4.
Response to Comments on the Quality of English Language
The english of this work is hard to read, and must be improved.
Response: We thank you for the comments on the manuscript English quality. We have tried to revised and refined the whole text.

Reviewer 3 Report
Comments and Suggestions for Authors
In the present manuscript, Karyasa et al. examined the development of hybrid biomaterials using silkworm cocoon filaments applying nano pastes of silica-phosphate-metal (Cu, Fe, or Al). The study is well-designed, and the findings are promising, especially in terms of antibacterial properties and color fastness. In my opinion, this is a valuable work and will be an interesting source for the scientific community. I suggest the following corrections:
1. Discussion part is very short. Provide more discussion of each result.
2. Be sure that mechanical and antibacterial results are validated statistically. I couldn’t see error bars and significance markers for all figures.
3. Indicate the environmental impact of the modified fibers.
4. Please add a brief discussion of potential areas of future research, e.g. testing silk fibers in industry, exploring different nano pastes, etc.
Comments on the Quality of English LanguageMinor editing of English is required. The text has typos. Please, correct them.
Author Response
Comments 1: Discussion part is very short. Provide more discussion of each result.
Response 1: Thank you for valuable suggestion. Accordingly, we provide additional discussion on effect of inorganic nano pastes of silicon-phosphate-Cu on the silkworm cultivation and the harvests on cocoon and silk fibers.
Comments 2: Be sure that mechanical and antibacterial results are validated statistically. I couldn’t see error bars and significance markers for all figures.
Response 2: We have, accordingly, elaborated and revised the Figure 6, Figure 10 and Figure 17.
Comments 3: Indicate the environmental impact of the modified fibers.
Response 3: Thank you. We add some environmental impact of the natural colored and siliconized, phosphorylated silk fibers and their production processes through feed engineering in silkworm cultivation. We proposed a discussion by using life cycle assessments beginning with mulberry cultivation, silkworm cultivation with feed engineering, cocoon processing and fiber reeling to silk yarn with in situ natural dyeing, and weaving the silk treads results into natural colored silk fabrics.
Comments 4: Please add a brief discussion of potential areas of future research, e.g. testing silk fibers in industry, exploring different nano pastes, etc.
Response 4: Thank you. We add a discussion about our future research on organic-inorganic hybrid materials based on the natural dyed silk fibers.
Response to Comments on the Quality of English Language
Minor editing of English is required. The text has typos. Please, correct them.
Response: We thank you for the comments on the manuscript English quality. We have tried to refined the whole text carefully to optimize the English grammar correctness as well as to minimize typos errors.

Round 2
Reviewer 2 Report
Comments and Suggestions for Authors
The original mechanical data, the stress-strain curves, must be provided in Figure 6 or in supporting information.
Comments on the Quality of English Languageit's ok
